# Fungal β-Glucan-Based Nanotherapeutics: From Fabrication to Application

**DOI:** 10.3390/jof9040475

**Published:** 2023-04-15

**Authors:** Fan Yang, Peter Chi Keung Cheung

**Affiliations:** School of Life Sciences, The Chinese University of Hong Kong, Shatin, New Territories, Hong Kong 999077, China; yeungf1018@gmail.com

**Keywords:** fungal β-glucans, nanomaterial, drug delivery, anti-cancer, anti-inflammation, immunomodulation

## Abstract

Fungal β-glucans are naturally occurring active macromolecules used in food and medicine due to their wide range of biological activities and positive health benefits. Significant research efforts have been devoted over the past decade to producing fungal β-glucan-based nanomaterials and promoting their uses in numerous fields, including biomedicine. Herein, this review offers an up-to-date report on the synthetic strategies of common fungal β-glucan-based nanomaterials and preparation methods such as nanoprecipitation and emulsification. In addition, we highlight current examples of fungal β-glucan-based theranostic nanosystems and their prospective use for drug delivery and treatment in anti-cancer, vaccination, as well as anti-inflammatory treatments. It is anticipated that future advances in polysaccharide chemistry and nanotechnology will aid in the clinical translation of fungal β-glucan-based nanomaterials for the delivery of drugs and the treatment of illnesses.

## 1. Introduction

Fungi have a wide range of pharmacological properties, including anti-inflammatory, anti-tumor, and immuno-modulating effects. They are an appealing source of physiologically functional foods and drug precursors, promoting the use of fungi in the food and pharmaceutical industries. β-glucans are found ubiquitously in fungi, plants, and bacteria; they are composed of glucose monomers linked together by glycosidic bonds. There is mounting evidence that non-cellulosic β-glucans, particularly those generated from fungi and yeast, are beneficial to human health due to their wide range of useful properties, excellent biocompatibility, and outstanding safety profile [1,2]. Fungal β-glucans have backbones made of 1,3-linked D-glucopyranose residues as common elements. However, the degree of branching, polymer lengths, and tertiary structures of β-glucans from distinct sources may vary greatly. Yeast β-glucans have extra β-1,3 regions and β-1,6 side branches, whereas mushroom β-glucans have attached β-1,6-branching points [3]. Notably, these structural diversities may impact the immunogenicity of fungal β-glucans, and several studies have demonstrated a correlation between increased structural complexity and improved immune system control as well as anti-cancer efficacy. Higher molecular weight fractions may affect their anticancer effectiveness in various experimental models, and the molecular size of the β-glucan molecules is also strongly connected to the biological impact [4].

β-glucans as integral components of most fungi cell walls in human fungal pathogens, are also key pathogen-associated molecular patterns (PAMPs) that elicit host immunity upon fungal infection by stimulating innate immune cells including macrophages, neutrophils, monocytes, natural killer cells (NK cells), dendritic cells (DCs), T cells, as well as granulocytes by binding to different Pattern Recognition Receptors (PRRs) such as Dectin-1, complement receptor 3 (CR3), and Toll-like receptor 2/6(TLR2/6)[5,6,7,8]. A number of factors influence receptor recognition and binding including solubility, molecular weight, and structure of the fungal β-glucans [9]. Of all the β-glucans from various sources, those originating from mushrooms and yeast have shown to possess remarkable anti-tumor and immune-modulating properties. It should be noted that instead of exerting a direct lethal impact on cancer cells or tumors, β-glucans stimulate immune cells and regulate the tumor microenvironment (TME), leading to a significant decrease in the development of the main tumor and distant metastases [10]. They are potential immune stimulants and have the ability to modify innate and adaptive immune responses inside the TME and enhance existing cancer immunotherapies [11,12,13,14]. Emerging research also centers on the anti-inflammatory effects of yeast, lentinan, and *Pleurotus* β-glucans [15]. In the last decade, the mechanisms on how β-glucan exerts anti-inflammation effects has been revealed by numerous scientific groups. According to a recent study, β-glucans stimulated innate immune memory, which supported the potential therapeutic utility of β-glucans in NLRP3-driven disorders by inhibiting NLRP3 activation and repressing IL-1β-mediated inflammation [16]. Another study indicated that nondigestible β-glucans extracted from *Grifola frondosa* blocked TLR2 receptors rather than Dectin-1 or CR3 receptors to exert anti-inflammatory effects [17]. It has also been discovered that oral consumption of β-glucan can reduce gut inflammation by controlling the NF-κB signaling pathway and altering the gut flora [18].

As a result, the therapeutic efficacy of β-glucans has been examined in over 200 clinical studies for a range of illnesses, including cancer, inflammation, obesity, cardiometabolic disorders, etc. For the last three decades, a soluble β-glucan medication produced from *Schizophyllum commune* has been authorized in Japan as a radiation enhancer for patients with cervical cancer [19]. Given the possibility that fungal β-glucans have great therapeutic potential, administering them along with conventional cancer therapies has helped patients recover by reducing the negative effects of chemotherapy and radiation [20]. However, it is challenging to translate any nutritional immunomodulatory agent like ß-glucans to produce synergies with chemotherapy drugs due to their rapid blood clearance, low-dose treatment, poor stability, etc. [21]. It was discovered that β-glucan is hardly ever detected in volunteers’ sera at any given time point, and that cytokine production and leukocyte microbicidal activity are not affected by orally administered yeast β-glucans [22]. The failure of clinical trials may be attributable to the administered β-glucans’ high viscosity, instability, low solubility, short half-lives, and low permeability into cells, which result in an extremely low blood concentration that is incapable of inducing an immunological response. Therefore, there is a great need to design β-glucan-based therapeutics to overcome these drawbacks.

Advancements in the development of nanotherapeutics has provided new therapeutic strategies for use as efficient vehicles for targeted drug delivery, controlled drug release, and gene therapy. It has been proven that the absorption of nanomaterials mainly depends on their particle size, with smaller particles being more easily absorbed than larger particles [23]. Nanomaterials with sizes ranging from 10 to 500 nm, such as nanoparticles, micelles, nanocapsules, etc., are able to improve the therapeutic efficacy and reduce the toxicity of therapeutic drugs by entrapping or conjugating them to form stabilized nanomedicines. These nanomedicines can be fabricated by manipulating their composition, structure, and function [24]. Compared to metal or inorganic nanomaterials, naturally occurred molecular building components including carbohydrates are more likely to be compatible with the biosystems inside the human body [25,26]. Therefore, it is worthwhile to investigate the potential of fungal β-glucans as nanotherapeutics to obtain improved physiochemical and immunomodulatory characteristics. As mentioned above, fungal β-glucans have a greater immunological potentiating effect on immune cells compared to other polysaccharides, giving them tremendous potential to be developed into nanotherapeutics rather than merely serving as drug delivery vehicles. Notably, β-glucan has been proven to be biosafe and non-toxic, lacking protein and peptide components that would trigger non-specific immune activation [27]. The US Food and Drug Administration (FDA) and the European Food Safety Authority (EFSA) have both recognized certain sources and formulations of fungal β-glucans as new food ingredients and dietary supplements [28]. Evidence has accumulated showing that large doses of fungal β-glucans up to 1500 mg per day increased plasma IL-10 and IL-10 mRNA in adipose tissue with no negative effects being noticed [28]. In addition, fungal β-glucans consist of a large amount of hydroxyl groups as the main active reaction sites for structural modification through sulfation, carboxymethylation, acetylation, hydroxyethylation, hydroxypropylation, methylation, amination, and esterification [29,30,31]. These characteristics allow them to be constructed into β-glucan-based nanoparticle systems with enhanced physiological properties and a high capacity for loading immune-modulating agents.

Current studies on polysaccharide-based nanomaterials are more focused on both well and lesser known polysaccharides with more functional groups and distinctive structures, such as chitosan, hyaluronic acid, cyclodextrin, starch, etc. [32,33], while β-glucans have been less thoroughly studied and receive little attention. Therefore, this review highlights synthetic strategies for fungal β-glucan-based nanomaterials and their use as nanotherapeutics and drug delivery systems for the targeted treatment of various diseases.

## 2. Design of Fungal β-Glucan-Based Nanotherapeutics

Most functional fungal β-glucans are water-soluble; hence, they can be extracted by water extraction aided by physical techniques such as heating, ultrasound, microwave, etc. [34]. Even though most fungal β-glucans are polar molecules that are readily soluble in water or alkaline solutions, efficient extraction of many insoluble glucans requires longer extraction times, higher pressures, and higher temperatures [1]. When using fungal β-glucans as nanotherapeutics or drug delivery vehicles, design tactics should be in line with their chain conformation and molecular weight, as well as the targeted application and its requirements, as these factors have a significant impact on their biological activities [35,36]. In contrast to other kinds of synthetic polymers and polysaccharides, certain fungal β-glucans have self-assembling properties in solutions and a reversible conformational transition between single- and triple-stranded helices, which enables them to bind specific polynucleotides and incorporate other guest molecules into the helix structure [37,38,39]. Additionally, chemical modification such as grafting positively charged groups or polymers is usually performed with the hydroxyl groups especially at the C6 position on the glucose unit due to the higher reactivity of this position [40]. In order to improve the biocompatibility of organic and inorganic nanomaterials, β-glucans are widely utilized as stabilizers [41,42,43]. In this section, we focus on optimal strategies to prepare nano-sized fungal β-glucan particles which have potential for theranostic applications (Figure 1).

### 2.1. Triple Helix

β-glucans can be found in solution in a wide variety of chain conformations, including coiled, single, duplex, and triplex chains, as well as spherical forms [44]. It is worthwhile to note that β-glucans from certain resources can form a triple-helix conformation in water which can be converted into a single random coil in DMSO or alkali solution. The formation and denaturation of the triple-helix structure depend heavily on the inter-hydrogen and intra-hydrogen bond interactions. The hydrophobicity of β-glucans due to the presence of hydrophobic carbon rings, contributes to their tendency to self-aggregate [45,46]. On one hand, β-glucans with triple helical structures can serve as chiral carriers to encapsulate certain polynucleotides, specific homopolynucleotides, such as polydeoxyadenosine or polycytosine, and diverse guest nanomaterials or polymers via hydrogen bonding [47,48]. On the other hand, they can also release those contents in some specific physiochemical environments according to the unique or chain behaviors of the triple helix [49]. Recently, studies have focused more on enhancing the binding stability and achieving appropriate particle sizes between triple-helix β-glucans and trapped molecules, especially polynucleotides, to meet the requirement of the FDA for nanomaterial utilization in drug products [50]. This can be achieved by modification of the β-glucans or polynucleotides.

Schizophyllan β-glucan (SPG), the most frequently investigated triple-helix β-glucan produced from *Schizophyllum commune*, is made up of a linear β-(1,3)-D-glucan backbone branched with β-(1,6)-D-glucose residues with various degrees of branching. Numerous studies have previously described the use of triple-helix SPG (tSPG) to combine CpG oligodeoxynucleotide (ODN) and small interfering RNA [51,52], and it appears that the molecular weight of SPG has a significant role in the binding affinity and stability of polynucleotides and SPG. Sumiya et al. [53] found that oligo-deoxyadenylic acid (dA_X_) could selectively bind to SPG with 30, 60, or 90 main-chain glucoses. These compounds displayed reduced particle sizes and could be better defined for the transition from therapeutic candidates to clinical trials. The molecular weight of SPG has an impact on the particle size of the complexes, which can also, in turn, affect their bioavailability. One study designed a CpG-binding SPG (CpG/SPG) in a nanogel with a larger particle size by producing cross-links between CpG/SPG molecules. This led to increased absorption of the complexes by antigen-presenting cells (APCs), which, in turn, activated the downstream immune responses [54]. Apart from gene delivery, some functional antigenic peptides can also be loaded on SPG. Mochizuki et al. [55] prepared SPG nanocomplexes for specific delivery of antigenic ovalbumin peptides and oligonucleotides to APCs, which prevents ovalbumin from being easily excreted in urine after administration into the body.

Lentinan (LNT) from the mushroom *Lentinus edodes* has two side chains regularly distributed on every five linear β-(1,3) glucose residues. Similarly, it has also been used in delivery systems for genes [56] and inorganic materials [48]. Triple-helix LNT β-glucans not only served as reducing agents for gold nanoparticles fabrication, but can also be subjected to denaturation and renaturation in order to prevent silver–gold nanoparticles (Ag-AuNPs) from aggregating over time due to the hydrophobic interaction between the hydrophobic cavity and the nanoparticles [57].

Yeast β-glucans (YSGs) from *Saccharomyces cerevisiae* prefer to adopt the shape of micro-nano particles in contrast to the triple-helix β-glucans mentioned, although they are still reported to have triple-helix structures [58]. As oral delivery vehicles for bioactives, microparticulate YSGs have attracted growing interest, although the full potential of nano-sized YSGs have not yet been realized. A recent work attempted to create unique chiral nanocarriers with a particle size of roughly 100 nm by employing the self-assembly of YSGs in a DMSO/water solution in the presence of Doxorubicin (DOX) [59]. Moreover, YSGs have also been formylated and self-assembled into rod-like, sheet-like, or tubelike supramolecular nanostructures by annealing in formic acid [60].

### 2.2. Covalent Cross-Linking

When encapsulating bioactive drugs in nanosized delivery systems, it is typically found that the structures of nanoparticles degrade, and the main chains of fungal β-glucans separate from the loaded agents before they reach the target sites. This can cause the loaded agents to be released prematurely, which can eventually cause systemic toxicity to healthy organs and reduce the effectiveness of the therapeutic intervention. One method for maintaining structural stability of fungal β-glucan-based nanocomposites is to introduce chemical cross-linking agents with at least two reactive functional groups that allow the formation of bridges between the bioactive agents/small molecules and fungal β-glucans. Typically, the chemical links in a network structure are designed to be cleavable or stimuli-responsive in order to release the payloads in response to environmental factors including low pH and reducing conditions [26,61,62,63].For example, Miao et al. [64] introduced a disulfide-containing cross-linker (2,2′-dithiolethanol) between YSGs and drugs, which enabled this prodrug to self-assemble into nanoparticles in aqueous environments, with hydrophilic β-glucans forming the outer shell and the linked drugs entrapped inside the inner core by hydrophobic interactions. Glutathione (GSH) is overexpressed in tumor microenvironment, which can specifically cleave the disulfide bonds of the prodrug and achieve a tumor targeting effect. Chen et al. [65] introduced 3,3′-dithiodipropionic acid (DTPA) to crosslink Methoxy poly (ethylene glycol) (mPEG) with YSGs to create self-assembling nanoparticles. These nanoparticles can encapsulate hydrophobic methotrexate (MTX) and release MTX after the reduction of the disulfide bonds. Schiff base linking is another common linking bond because the terminal hydroxyl groups on fungal β-glucans are easily oxidized into aldehyde groups [66]. As a result of the Schiff bond’s high sensitivity to pH and tendency to hydrolyze at low pH levels, bioactive agents can be released more readily in the slightly acidic tumor or inflammation environment. DOX with amino groups can conjugate with a oxidized LNT β-glucan (LNT-CHO) through the Schiff base reaction [67]. Poly(ethylenimine) (PEI) with cationic amino groups is also popular as a fungal β-glucan-conjugate since it can also be used for negatively charged CpG ODN loading and can bind to cell membranes by electrostatic interactions [10].

### 2.3. Electrostatic Interactions

Much emphasis has been paid to the utilization of electrostatic interactions between fungal β-glucans and counterions or polyelectrolytes. This simple and mild method offers several distinctive benefits, including a non-toxic procedure, reversible binding, a process devoid of organic solvents, and simple scaling. Sometimes nanosystems prepared by ionic complexation does less harm to the protein structure than chemical cross-linkers. The materials used to create nanocomplexes are macromolecules with opposing charges. Since fungal β-glucans are mostly negatively charged, they can be decorated onto positively charged chitosan that consists of amino groups. Chitosan is a frequently used polysaccharide drug delivery nanoplatform but lacks unique targeting moieties, and this problem can be solved by coating β-glucans onto the surface to enhanced cancer cell therapy [68]. To offer the reaction site for electrostatic interactions, hydrogen groups in fungal β-glucans are frequently converted into functional groups with a particular charge by chemical modification. For example, YSGs were modified with carboxymethyl groups and connected with cationic chitosan polysaccharide to form extremely stable antigen vehicles for ovalbumin delivery [69]. LNT β-glucans can also be carboxymethylated into polyanionic electrolytes to be electrostatically integrated with positive charged poly-l-ornithine (PLO), which can strongly improve the biocompatibility of the drug delivery system [70]. Apart from carboxymethylation, amination is frequently conducted to produce glucans with cationic properties. Aminated-β-glucans with enhanced water solubility were used to encapsulate CpG ODN by electrostatic interactions [71]. Cationic β-glucans can also be prepared using 3-chloro-2-hydroxypropyltrimethylammonium chloride (CHPTAC) as the etherifying agent [72].

### 2.4. Stabilization of Metal/Inorganic Materials

Another strategy applied in recent years is using fungal β-glucans as stabilizer components to develop organic–inorganic hybrid nanocomplexes. Fungal β-glucans can serve as an effective template for nucleation and help synthesize inorganic nanoparticles by decreasing the corresponding metal salts. Typically, metal–polymer coordination can create stronger bridges between β-glucans chains by chelation between negatively charged ligand moieties of β-glucans and metal cations, as opposed to covalent cross-linking. Li et al. [73,74] used mushroom β-glucans extracted from *Pleurotus tuber-regium* (PTR) to enhance the stability and biocompatibility of gold nanorods through electrostatic interactions as the zeta potential of the complexes shifted from a positive to negative charge. However, there are also studies that demonstrated that β-glucans can stabilize metal nanoparticles, such as gold nanoparticles, due to the strong interaction of Au-O bonds between gold and the hydroxyl groups on glucans [75]. Recently, it was reported that metal ions will bind with sulfur-containing groups of organic components and transform into solid minerals with no heating process required [42,76]. Su et al. [77] synthesized carboxymethylated β-glucans extracted from mushroom sclerotium of PTR to stabilize iron oxide nanoparticles by electrostatic interactions and strong Fe-O binding. Apart from metallic materials, non-metallic materials such as selenium nanoparticles (SeNPs) are also popular as a potential therapeutic drugs for cancers. Unmodified SeNPs are unstable and prone to aggregation, which hinders their biological applicability. Since fungal β-glucans have several hydroxyl groups and large surface areas, they are suitable for decorating SeNPs and forming hydrogen bonding connections between the hydroxyl groups on the β-glucans chain and selenium [78,79,80].

### 2.5. Emulsification

The emulsion-based method is commonly used to prepare polymer nanoparticles. Nanoemulsions exhibit high kinetic and thermodynamic stability, which allows them to be functionalize for applications in materials science and biotechnology. The emulsified system is generated by combining two immiscible liquids (water and oil) to form a single metastable phase using an appropriate surfactant or its mixture, and mechanical processes. There are two types of methods for the mechanical processes, one of which is called the high-energy methods. For example, vortexing or stirring, homogenization [81], high-shear mixing [82], high-pressure microfluidics [83], and ultrasonification [84] all offer the required mechanical energy to overcome the substantial interfacial energy barrier to produce dispersions of nanoscale droplets. The other type is low-energy emulsification that only needs tiny amounts of external energy such as changing the surfactant concentration or temperature for droplet formation [85]. Emulsions are often divided into two categories: oil-in-water (O/W) direct emulsions and water-in-oil (W/O) inverse emulsions, where water or oil is the continuous phase, respectively. Moreover, multiple emulsions such as oil-in-water-in-oil or water-in-oil-in-water can also be formed to encapsulate drugs with different solubilities into various phases.

The use of polysaccharides, especially fungal β-glucans as emulsifiers, has been widely studied because it can achieve the purpose of reducing the interfacial tension between the water phase and the lipid phase, thereby reducing the size of the droplets. Furthermore, β-glucans have the potential to increase polymer viscosity and promote depletion flocculation to reduce lipid digestion in the small intestine [83]. In addition to their function as emulsifiers, β-glucans are commonly utilized in the creation of nanoemulsions as drug delivery systems or as loading agents [86]. Due to the stability of the delivery system for a multiple-nanoemulsion system and the unique triple-helix structure, they have been demonstrated to have longer pharmacokinetic lifetimes and lower dosage requirements for both hydrophobic and hydrophilic drugs [87,88,89,90].

Interestingly, β-glucan nanoemulsions can be rapidly and flexibly formed to produce spherical β-glucan structures due to efficient supramolecular wrapping in the aqueous phase of the O/W emulsion interface [91]. Numata et al. [92] created giant polymer micelles with a uniform diameter around 200 by mixing poly(styrene) (PS) with SPG, with the β-glucan fastening the PS chains together in a noncovalent fashion to facilitate the formation of a supramolecular polymer network on the O/W emulsion surface. However, all the above nanoemulsion formations require a large number of surfactants. There is an emerging novel technique for producing nanoemulsions using solid particles such as PLGA to replace the surfactants to form the so-called pickering emulsions, which can improve the stability, biocompatibility, and bioavailability of the emulsion [93]. Jiao et al. [94] also used PLGA solid nanoparticles as stabilizers to prepare LNT/PLGA pickering emulsions for an ovalbumin loading system and activation of immune responses.

### 2.6. Nanoprecipitation

Nanoprecipitation, also known as desolvation, differs from emulsion-based approaches mainly because it takes place in a miscible solvent without using any surfactants [95]. This approach often involves coacervating or precipitating a modified β-glucan matrix in an organic, polar, and water-miscible solvent. Due to the reduction in their solubilities in the aqueous medium, solute particles precipitate out. Following the displacement or removal of the organic solvent, spherical β-glucans nanoparticles suspensions are frequently produced with a diameter range from 20 to 500 nm. The methods for removing solvents include lyophilization, dialysis, and evaporation. Compared to other fabrication methods, nanoprecipitation is a simple one-step method and does not require any extra heating [96]. The size and polydispersity (PDI) of β-glucan nanoparticles may be controlled by several experimental factors, such as the two phase mix, rate of solvent addition to a non-solvent, solute concentration, etc. As such, nanoprecipitation must be carefully managed because it could also have an impact on the stability of the final nanoparticles. As a simple and rapid method, it has been used to prepare drug-loaded fungal β-glucans nanoparticles. Suo et al. [97] prepared drug–drug nanoprecipitation products simply by dissolving LNT β-glucans and the poorly water-soluble drug Regorafenib in DMSO, then added the mixture solution into water with stirring. These nanosuspensions were highly stable due to the steric hindrance formed by LNT β-glucans on the drug surface to prevent the drug from aggregating. Shah et al. [98] dissolved rutin in ethanol and added it dropwise into a mushroom β-glucan solution under stirring followed by ultrasonication.

Nanoparticles constructed with amphiphilic diblock copolymers are often stabilized in aqueous solution. By introducing hydrophobic molecules into the hydrophilic β-glucans backbones, amphiphilic block β-glucans nanoparticles prepared by nanoprecipitation typically exhibit a spherical core composed of hydrophobic blocks surrounding by a shell of hydrophilic blocks. The degree of substitution (DS) of hydrophobic molecules has a great impact on the physicochemical properties of the nanoparticles, such as their size, surface characteristics, and shape. It was reported that carbohydrates with low DS values often fail to form self-assembled nanoparticles, but excessive modification of carbohydrates may precipitate under aqueous circumstances due to their high hydrophobic interactions [99]. DS sometimes has a substantial impact on the biological activities of the parent β-glucans. In addition, the ratio of organic phase and water phase are decisive for the size of the formed nanoparticles. Hence, it is crucial to determine the best DS value for each hydrophobic molecule and miscible phase ratio when performing nanoprecipitation. When using a ethanol to water ratio of 10:1, *Ganoderma lucidum* nanoparticles without any modification were fabricated with the smallest size around 95 nm and highest stability [100]. Numerous hydrophobic moieties can be grafted onto β-glucans such as ester groups [101,102] and alkyl groups [101]. Wu et al. [101] grafted hydrophobic ester groups on YSG hydroxyl groups to synthesize self-assembling nanoparticles via nanoprecipitation, with glucan esters dissolved in acetone followed by dialysis against distilled water. The β-glucan ester with highest DS had the smallest sizes of around 132 nm, which can be potentially developed as drug delivery vehicles. YSGs conjugated with butyryl groups could self-assemble into nanoparticles and function as DOX carriers by optimizing the DS of the butyryl groups and DMSO/water ratio [59].

### 2.7. Other Techniques

There have been various mechanical processes that were utilized to generate β-glucan nanoparticles in addition to the main methods discussed above. Mechanical techniques are regarded as green technology since they do not require additional chemicals and may maintain the fundamental structure and natural characteristics of the β-glucans. Ultrasonication, which uses high-intensity sound waves to reduce the particle size, is commonly employed in particle size reduction. Periodic waves are produced during ultrasonic irradiation, which agitates the material and causes bubbles to burst. This causes high temperatures and pressures to break down the macromolecular polymer chains [103,104]. By using an ultrasonic cell disruptor, micron-sized yeast cells could be reduced to three distinct sizes of nanoparticles (50, 200, and 500 nm), and they showed high anticancer and immunotherapy efficacy [105]. Controlling the ultrasonic period is crucial because studies have shown that although short-duration ultrasound creates thin particles, long-duration ultrasound causes reaggregation and destroys the structure of the glucans [106]. Solvent-free ball milling is gaining popularity which is attributed to its ease of use, cheap cost, and environmental friendliness, as well as its capacity to obtain extremely high yields. Metal balls, such as zirconia (ZrO_2_) or steel balls, serve as a grinding medium and spinning shells to generate centrifugal forces, and are the essential components of a shear-force dominated process where the particle size continues to be reduced by impact and attrition [107].

## 3. Applications

Fungal β-glucans represent the most widespread biomolecules in nature. They possess many beneficial properties that make them promising and excellent candidates in the development of biomaterials and even theranostic nanomedicine as they reduce concerns regarding biocompatibility, biodegradability, toxicity, and physiological stability (Table 1) [108,109]. The stability of drugs combining with fungal β-glucans can be better maintained since fungal β-glucans with branches attached to the outer layer may provide steric hindrance and electrostatic repulsion. This allows the duration of drug treatment to be extended, and their structural integrity can be maintained by their dynamic state [110]. For oral administration, fungal β-glucans are non-digestible fibers that may escape quick digestion brought on by endogenous mammalian enzymes in the gastrointestinal tract (GIT), which can overcome shortcomings such as low drug absorption, brief residence period at the disease site, and drug instability. As a result, fungal β-glucans are being explored as potential therapies for metabolic syndrome, obesity, and diet management due to their impact on the gut flora, lipid and glucose metabolism, and cholesterol levels [111,112,113]. Meanwhile, for systemic administration, the majority of negatively charged fungal β-glucans may expedite blood circulation and circumvent the reticuloendothelial system. Additionally, fungal β-glucans interact with particular receptors to exhibit a variety of biological functions, which extends the in vivo residence period of nanoparticles and raises the potential for disease site accumulation. The aforementioned benefits significantly improve physical stability and boost theranostic function in vivo, both of which indicate promising application potential.

### 3.1. Anti-Cancer

Cancer continues to be one of the top causes of mortality globally, and the development of more effective treatments remains a challenge for scientists. High incidence and fatality rates related to cancer are mostly attributed to the advanced disease’s resistance to traditional therapies including surgical resection, chemotherapy, and radiotherapy [114]. In this case, nanotechnology has the potential to play a role in cancer therapy with fewer side effects and avoiding high risks of malignant tumors, cancer metastasis, and recurrence in order to overcome the limitations of conventional therapies. Well-designed nanoplatforms with the ideal size and surface qualities can be accumulated in the tumor tissue with improved therapeutic effectiveness and fewer side effects by taking advantage of the passively increased permeability and retention (EPR) effect and conjugating targeted molecular moieties on the nanoplatform. Fungal β-glucan nanoparticles are regarded as one of the best drug delivery systems/nanodrugs to treat cancer because of their excellent biocompatibility and the availability of multifunctional conjugation. Extensive studies have involved fungal-derived β-glucans for cancer treatment [115]. Fungal β-glucans have four main functions in cancer nanotherapeutics: (1) encapsulated nanocarriers; (2) targeting of ligands for specific cell interactions; (3) surface decorator to improve structure stability; and (4) immunity regulators for cancer immunotherapy.

Recent studies have demonstrated that β-glucan nanoparticles increased humoral immunity by impacting the innate and adaptive immune systems via immune cells such as antigen-presenting cells like dendritic cells and macrophages, T cells, and B cells. β-glucans extracted from *Ganoderma lucidum* were used as an immunomodulatory nanoplatform to load with chemotherapy drugs such as DOX [42], or to coordinate with radiotherapy [66]. These nanoplatforms can specifically target DCs, produce immature DCs, and subsequently activate DC-derived T lymphocytes, which have a potent inhibitory effect on tumor development and metastasis. Targeting M2-like Tumor-associated macrophages (TAMs) or educating TAMs to become M1-like macrophages have emerged as promising therapeutic approaches, since M2-like macrophages can enhance tissue remodeling and block anti-tumor T cell responses [116]. Carboxymethylated fungal β-glucans can serve as a macrophage-targeting transporter for iron oxide nanoparticles, therefore inhibiting tumor development by promoting the polarization of macrophages into proinflammatory M1 macrophages. The underlying mechanisms are related to mitogen-activated protein kinase (MAPK) activation and the tyrosine kinase (Syk)/nuclear factor kappa-B (NF-κB) signaling pathways [77,117,118].

When it comes to solid tumors, tumor-draining lymph nodes (TDLN) are believed to be involved in tumor progression. Directing the treatment to the TDLN might potentially take advantage of anti-tumor T cell production and have a beneficial effect on tumor growth control [119]. Nevertheless, the size of the treatment platform affects uptake efficacy [120], as nanoparticles smaller than 200 nm can readily enter the lymphatic system by fenestrated lymph vessels and reach subcapsular sinuses within minutes, but bigger nanoparticles can only be delivered by DCs and reach T cell zones in lymph nodes after 24 h [121]. Xu et al. [105] utilized yeast that contained about 88.20% β-glucans, to fabricate yeast-based nanoparticles (YCW NPs) with small (~50 nm), middle (~200 nm), and large (~500 nm) sizes. They reported that the smallest YCW NPs were found to not only exert the most remarkable effect on reversing the immune-suppressive microenvironment in tumors to inhibit tumor growth, but they also targeted and penetrated into TDLNs to prompt the maturation of DCs and activation of T cells and B cells.

### 3.2. Nanovaccines

Due to its natural PAMP properties and immunoregulatory effects, fungal β-glucans have been applied in the development of nanovaccines and gene therapy [122]. It is reported that vaccines made of fungal β-glucans can cause trimethylation of histone H3 at Lys4 (H3K4me3) and acetylation of histone H3 at Lys27 (H3K27ac) in monocytes and macrophages [5]. These modifications persisted for several weeks after the stimuli were removed, resulting in an enhanced epigenetic state to reprogram the epigenetic landscape of innate immune memory. Current efforts towards improving vaccine efficacy focus on employing nanoparticles for the delivery of genes or antigens. Studies have demonstrated fungal β-glucan-based nanoparticles were helpful in the creation of effective and safe vaccines since it can enhance the immune response, minimize adverse effects, accelerate immunomodulatory activity, and prolong antigen and gene residency [123]. The triple-helix structure of β-glucans extracted from *Schizophyllan* has been a promising antigen-binding source in this field [51,52,54,55,124]. YSGs particles as vaccine delivery platforms were popular even in clinical trials due to their inexpensive cost, simplicity of culture, ease of scalability, and ability to induce robust humoral and cellular immune responses [125]. For example, a peptide antigen-linked β-glucan nonvaccine exert immunomodulatory effects via systemic administration [126]. In conclusion, fungal β-glucans may function as immunostimulators as well as carriers that enable the transfer of drugs, antibodies, or genes included in the conjugate to the lymph nodes and other immunological organs, where additional immune activation will occur.

### 3.3. Anti-Inflammation

Inflammation is the first biological response of the immune system that is triggered by noxious stimuli and conditions, such as infections and tissue injuries [127]. Inflammatory responses are triggered by microbial infections, leading to infiltration of inflammatory cells, excessive production of a variety of inflammatory mediators and reactive oxygen species, dysregulation of cellular signaling, and increasing permeability of endothelial lining cells. Excessive inflammatory responses could result in persistent inflammation, which is linked to a wide variety of diseases including cancer. Fungal β-glucans were reported to possess anti-inflammatory effects [128]. However, the mechanisms of how β-glucans influence human health have not been unveiled. Some of the fungal β-glucans were found to have the potential to decrease the expression of inflammatory factors such as TNF-α, IL-1, IL-6, IL-8, and MCP-1 by regulating several signaling pathways in immune cells especially macrophages [129,130]. At an inflamed location, activated macrophages play a leading role in driving the inflammatory cascade that cytokines and chemokines set in motion. Since β-glucans can be recognized by the dectin-1 receptor and CR3 on macrophages, β-glucan nanoparticles can exhibit immunomodulatory and anti-inflammatory effects by delivering bioactive components precisely to remote inflammatory sites via intricate macrophage-mediated transportation [58,65]. In addition, numerous studies have recently been conducted to determine how fungi and β-glucans affect the activation of inflammasomes and the consequent generation of IL-1β. Mice with defective NLRP3 exhibit higher fungal loads and reduced resistance to infections with *Candida albicans*, *Acinetobacter fumigatus*, or *Cryptococcus neoformans* [131]. β-glucans extracted from *Pleurotus albidus* had a substantial impact on the NLRP3 inflammasome in macrophages, which improved macrophage performance in order to reduce lipid-induced inflammation and foam cell formation [132].

**Table 1 jof-09-00475-t001:** Examples of β-glucan from different fungi resources used as nanotherapeutics.

**Source**	**Modification and Fabrication Strategy**	**Benefits**	**Application**	**References**
*Lentinus edodes*	Triple helical LNT β-glucan conjugated with poly(dA)–SS–PS-ATNF-α	Improve thermal stability under physiological conditions, protect poly(dA) from degradation	Intestinal Inflammation therapy	[56]
	Oxidized LNT (LNT-CHO) conjugated with DOX via Schiff base	Enhance the cytotoxicity effects against breast cancer cells	Cancer therapy	[67]
	Carboxymethyl LNT coated on mesoporous silica nanoparticles for DOX loading	Improve cytocompatibility, hemocompatibility, and histocompatibility	Cancer therapy	[70]
	LNT coated on selenium nanoparticles (SeNPs)	Prevent nanoparticle from aggregating and improve treatment efficacy	Cancer therapy	[78]
	LNT as stabilizer during SeNP synthesis	Improve stability and target to toll-like receptor-4 (TLR4)	Anti-inflammation	[80]
	Synthesize LNT-Regorafenib (RG) nanosuspension by nanoprecipitation	Improve the oral bioavailability of RG	Cancer therapy	[97]
*Schizophyllum commune*	Triple helical SPG-encapsulated siCD40	Target immune-regulating monocytes and DCs dectin-1 receptors	Dectin-1-targeted delivery	[51]
	Synthesize ODN/SPG nanogel particle using cross-links between CpG/SPG and complementary CpG/SPG complexes through DNA–DNA hybridization	Provide a suitable delivery vehicle to transport the CpG-ODNs to APCs	Vaccine adjuvant	[54]
	Triple helical SPG for peptide-dA/SPG and CpG-dA/SPG complex synthesis	Promote complex accumulation in immune tissues	Vaccine adjuvant	[55]
*Pleurotus*break/> *tuber-regium* (PTR)	PTR β-glucans coated on the surface of gold nanorods (AuNRs)	Enhance stability and overcome the cytotoxicity of AuNRs for cancer photothermal therapy	Cancer photothermal therapy	[73,74]
	Carboxymethylated PTR β-glucans coated on iron oxide nanoparticles (IONPs)	Target macrophage and trigger immune responses in combination with IONPs	Cancer immunotherapy	[77,118]
Yeast	Denaturing/renaturing of YSGs in the presence of DOX to form complexes	Strong immunopotentiation ability and high drug loading efficiency	Cancer therapy	[59]
	Amphiphilic YSG conjugates via grafting water-soluble PEG through esterification for MTX loading	GSH-responsive release of MTX, target macrophages and drive macrophage polarization	Rheumatoid arthritis therapy	[65]
	Anionic carboxymethyl-YSGs conjugated with cationic chitosan and OVA via electrostatic interaction	Facilitate the recognition by APCs	Vaccine adjuvant	[69]
	Yeast broken down by ultrasonic cell disruptor to prepare YSG nanoparticles with different sizes	YSG nanoparticles with smallest size accumulate in tumor-draining lymph node (TDLN), remodel the immunosuppressive microenvironment in tumors and TDLNs	Cancer immunotherapy	[105]
	Nanoparticle reduction of YSGs by ball milling technique	High solubility and bioavailability	Nutraceutical potential	[107]
*Ganoderma lucidum* (GLP)	Thiol end-functionalized GLP conjugated with bismuth sulfide nanoparticles (BiNP) through metal–sulfur bond	Enhance immune response and protect the kidney that may be damaged by large doses of bismuth	Cancer radiotherapy	[66]
*Agaricus bisporus*	*Agaricus bisporus* β-glucans encapsulation of rutin by nanoprecipitation	Protect complexes from the harsh gastrointestinal environment and to enhance bioavailability	Anti-obesity and antioxidant activities	[98]

## 4. Conclusions and Perspective

Fungal β-glucans have garnered considerable attention for biomedical applications in the past decade mainly because of their wide variety of biological activities. The employed fungal β-glucans are derived from a variety of natural sources, making them an ideal choice for sustainable nanotechnology. There is a wide range of synthetic strategies for nanoparticles containing fungal β-glucans and their derivatives. According to previous studies, the conformation, molecular weight, presence of particular functional groups, and degree of branching of fungal β-glucans may be associated with their abilities to stimulate the immune system. Meanwhile, to fabricate excellent fungal β-glucan-based nanocarriers/nanoparticles for in vivo delivery systems, controllable size, charge, surface morphology, loading capacity, and targeted release of substances are crucial. The choice of an appropriate synthetic method, such as triple-helix conformation, emulsification, nanoprecipitation, complex coacervation, or other appropriate methods, is contingent on the particular application and its needs. As effective nano-sized carriers or medicines, fungal β-glucans exert these bioactivities by binding with cell surface receptors and triggering subsequent signaling cascades. Moreover, through modifying metabolism, fungal β-glucan nanoparticles might retard aging, exhibit antioxidant abilities, postpone skin aging, and function as probiotics. Although the immunomodulatory and anti-inflammatory activities of fungal β-glucans have been extensively studied, only a negligible amount of development can be translated into clinical studies due to the impurity of products and complex chemical conjugation, which result in unexpected toxicity after systemic administration. Furthermore, their disease-targeting ability, drug release, and degradation profile are also major issues. Future research on fungal β-glucan nanoparticles should aim to enhance the sensitivity and specificity of stimuli-responsive triggering, as well as the safety profile. To fully exploit the therapeutic potential of fungal β-glucan nanoparticles for human health and disease, a more systematic preparation technology as well as a quality assessment standard should be developed.

## Figures and Tables

**Figure 1 jof-09-00475-f001:**
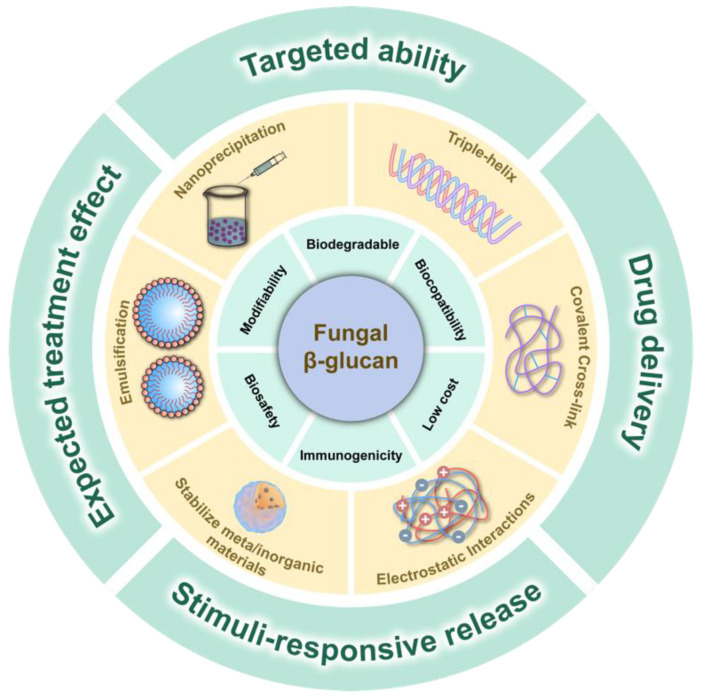
Schematic illustration of design and consideration of fungal β-glucan-based nanotherapeutics.

## Data Availability

Not applicable.

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
