# Peer review of "Fungal β-Glucan-Based Nanotherapeutics: From Fabrication to Application"

_jof, 2023, doi:10.3390/jof9040475_

Round 1

Reviewer 1 Report (Previous Reviewer 1)

The manuscript is well written and provided a comprehensive review of Fungal β-glucan-based Nanotherapeutics: From Fabrication to Application.

Author Response

Thanks for the reviewer's positive comments. 

Reviewer 2 Report (Previous Reviewer 4)

The review describes synthetic strategies of fungal beta-glucan-based nanomaterials and their use as nanotherapeutics and drug delivery systems for treatments of various diseases. The review well highlights how these molecules, known for their immunomodulatory properties and for their excellent biocompatibility and outstanding safety profile, are used as nanomaterial for the delivery of drugs and simultaneously as immunomodulators to facilitate the treatment of various illnesses. In particular, the review describes, according to the different characteristics of the beta glucans extracted from different mushrooms or yeasts, the various technological strategies that allow their use as carriers i.e. triple helix, covalent cross-link, electrostatic interactions, stabilize metal/inorganic materials, emulsifications, nanoprecipitations. Moreover, the review provides an updated and accurate description of the various testing preparations based on beta-glucan in use for anti-cancer therapies, vaccine formulations and treatment of some inflammatory pathologies.

The review is of interest to a large audience.

Minor points:

line 522 only shows the title of paragraph “6. Patents” without any description. It would be better to remove it.

Author Response

Thanks for the reviewer's suggestion. We removed "6. Patents" in our revised manuscript. 

This manuscript is a resubmission of an earlier submission. The following is a list of the peer review reports and author responses from that submission.

Round 1

Reviewer 1 Report

Abstract

Line 9: change β-glucans to β-glucan

Line 14: new developments o polysaccharide chemistry. Check typographical errors

Introduction

Line 41: Expand NK cells, DCs

Line 46: change 'the most' to remarkable

Include information on the extraction methods of β-glucan from fungal sources

How β-glucan is advantageous over other polysaccharides?

Provide information on the currently available β-glucan based nanoparticles

Include the mechanism of anticancer and antiinflammatory action of β-glucans

Reviewer 2 Report

Dear authors,

I would like to congratulate you on the valuable insights you provide in the broad field of fungal nano-therapeutics. The manuscript is scientifically sound and well written. However, it must be checked for a few errors [such as keyword 1 (line 17); of (line 15) etc.]. 

Also I would like to suggest to rewrite the abstract (lines 7 to 10) as it reads very broad and doesnot contribute towards the significant details provided in the manuscript.   

Keywords should also be rewritten (such as "application" should be omitted and specific application should be added)  

Many immunological cells are mentioned with an assumption of understanding about their short forms it is preferred to add their full forms at their first use in the manuscript. As I suppose this paper will have a wider audience. 

Thanks and Regards

Reviewer 3 Report

The target receptor for beta glucan from yeast and fungi is well known and represented in the literature as Dectin-1. There is significant literature supporting the immune response and cytokine production resulting from treatment of many different beta glucans of natural sources. Several statements regarding activity and toxicity are presented with out reference support. 

I have tried to highlight some places where grammar is an issue and sentences that don’t make sense. I feel this review should be rejected or re-done with a more directed focus on a disease area or SAR. Many of the topics are already recently covered in other reviews and it is unclear how this review expands this knowledge.

Line 50-56. All statements are basically the same and do not differentiate how the sources are different. This could be one sentence with multiple references

Line 58: is this really unknown? Beta glucan initiates IL-10 secretion, a know anti-inflammatory cytokine

Line 70-74: sentences do not make sense

Line 76: have any radio-label studies been done? These are sugars, of course they are easily digested.

Line 80: what are the drawbacks? One extract has been used clinically for 30 years. How did other products fail?

Line 83: spelling/grammar

Line 108: How? References needed and more explanation and it was stated in the introduction that chemistry would be discussed

Line 124: other polysaccharides can’t self-assemble? Inulin?

Line 127: is there a reference to support this?

Line 131: is there a reference to support this?

Line 137: by adding DMSO or does just dissolving in DMSO break the helix, does it need heat and base?

Line 141-143: and on the other hand? How does this review fill the gaps that the review Ref 37 did not. Is this review only covering the last two years of progress in beta glucan advances?

Line 256: is this a different reference that the Su paper?

Line 262: sentence is confusing

Line 512: we are just now being introduced to the toxicity of degrade glucans?

The conclusion states many issues that I don’t think were discussed anywhere else in the review and this is very confusing.

Reviewer 4 Report

The review is very interesting and describes synthetic strategies of fungal beta-glucan-based nanomaterials and their use as nanotherapeutics and drug delivery systems for the targeted treatments of various diseases. The review well highlights how these molecules, known for their immunomodulatory properties and for their excellent biocompatibility and outstanding safety profile, are used as nanomaterial for the delivery of drugs and simultaneously as immunomodulators to facilitate the treatment of various illnesses. In particular, the review describes, according to the different characteristics of the beta glucans extracted from different mushrooms or yeasts, the various technological strategies that allow their use as carriers of various therapeutic molecules i.e. triple helix, covalent cross-link, electrostatic interactions, stabilize metal/inorganic materials, emulsifications, nanoprecipitations. Moreover, the review provides an updated and complete picture of the various testing preparations based on beta-glucan in use for anti-cancer therapies, vaccine formulations and treatment of some inflammatory pathologies. For its clarity, the review is of interest to a large audience. 

Minor points:

line 520: the paragraph title 6. Patents should be erased